# Lipophilized Epigallocatechin Gallate Derivative Exerts Anti-Proliferation Efficacy through Induction of Cell Cycle Arrest and Apoptosis on DU145 Human Prostate Cancer Cells

**DOI:** 10.3390/nu12010092

**Published:** 2019-12-28

**Authors:** Jun Chen, Linli Zhang, Changhong Li, Ruochen Chen, Chengmei Liu, Mingshun Chen

**Affiliations:** State Key Laboratory of Food Science and Technology, Nanchang University, Nanchang 330047, China

**Keywords:** EGCG, LEGCG, DU145 cell, proliferation, apoptosis

## Abstract

Epigallocatechin gallate (EGCG) is the predominant tea polyphenol and it exhibits a hydrophilic character. The lipophilized EGCG derivative (LEGCG) was synthesized by enzymatic esterification of EGCG with lauric acid to enhance its bioactivity. The tetralauroyl EGCG was confirmed by high-performance liquid chromatography-tandem mass spectrometry and further identified as 3′, 5′, 3″, 5″-4-*O*-lauroyl EGCG by ^1^H and ^13^C nuclear magnetic resonance. The anti-proliferation effect of LEGCG on DU145 human prostate carcinoma cells was evaluated by MTT assay. In addition, the underlying molecular mechanism by which LEGCG exerts anti-proliferation efficacy was elucidated by flow cytometry and immunoblot analysis. Results suggested that LEGCG exhibited a dose-dependent anti-proliferation effect on DU145 cells by G0/G1 phase arrest and induction of apoptosis. LEGCG induced cell cycle arrest via p53/p21 activation, which down-regulated the cyclin D1 and CDK4 expression. In addition, LEGCG induced apoptosis by increasing the Bax/Bcl-2 ratio, the cytochrome *c* release, and the caspases cleavage on DU145 cells. The results provide theoretical support to prevent prostate cancer with LEGCG.

## 1. Introduction

Tea contains substantial amounts of polyphenols and has been shown to render multiple health benefits [1]. Epigallocatechin gallate (EGCG), the main tea polyphenol, shows several remarkable physiological activities, including antioxidant activities [2], antitumor activities [3,4], and other biological functions [5,6,7]. Because of its polyhydroxy structure, EGCG is highly hydrophilic and low lipophilic. The low lipophilicity of EGCG may limit its bioavailability because sufficient concentrations of EGCG is difficult to reach in vivo, which hampers its physiological activities.

Prostate cancer is a commonly diagnosed malignancy among males in the United States, and diet is considered a likely environmental risk factor [8]. As we know, there are huge differences in the diets between East Asia countries and Western countries; East Asia diets are dominated by soybeans, tea, fish, fruits, and vegetables, while Western diets are mainly focused on red meat and fatty foods [9]. The lower incidence of prostate cancer in East Asia countries than in Western countries may be attributed to the green tea polyphenols [10]. According to the literature, the intake of tea polyphenols prevents prostate carcinogenesis in vivo [11,12].

There are some reports about the synthesis of lipophilized EGCG derivative (LEGCG) with EGCG, and LEGCG shows good antioxidant activity [13,14]. But there are few articles on the anti-tumor activity of LEGCG [15]. In our previous study, lipophilic grape seed proanthocyanidin, synthesized by grape seed proanthocyanidin and lauric acid, showed excellent anti-prostate cancer affects [16]. Therefore, we hypothesized that LEGCG has the potential to prevent prostate cancer. In the present study, enzymatic esterification of EGCG was implemented. The high-performance liquid chromatography-tandem mass spectrometry (HPLC-MS/MS) and ^1^H and ^13^C nuclear magnetic resonance (NMR) were used to determine the chemical structures of LEGCG. The inhibitory effect and the underlying mechanism of LEGCG on DU145 cells were investigated by 3-(4,5-dimethyl-2-thiazolyl)-2,5-diphenyl-2-H-tetrazolium bromide (MTT) assay, flow cytometry, and immunoblot analysis.

## 2. Materials and Methods

### 2.1. Materials

EGCG, MTT, and propidium iodide (PI) were purchased from Sigma-Aldrich Co. (St. Louis, MO., USA). Annexin V-FITC/PI kit was from Biolegend (San Diego, CA, USA). Cyclin D1, CDK 4, p21, p27, p53, Bax, Bcl-2, cytochrome c, cleaved caspase-9, cleaved caspase-3, and GAPDH antibodies were from Cell Signaling Technology (Danvers, MA, USA).

### 2.2. Preparation of Crude EGCG Derivative

LEGCG was prepared according to the literature [17]. Briefly, EGCG and lauric acid were mixed at a mole ratio of 1:1 in ethanol, and then 5% of Lipozyme TLIM (5% *w/w*) was added. The mixture was heated at 45 °C for 12 h in a screw-capped glass bottle. The reaction was terminated by the removal of the enzyme through filter paper, and the product was concentrated to get crude LEGCG.

### 2.3. Purification and Identification of LEGCG

Waters HPLC systems comprise a 1525 pump and a 2998 diode array detector (DAD) was used for analysis. The separation was performed by a 5 μm Waters Atlantis T3-C18 column (250 × 4.6 mm). The mobile phase was 0.1% acetic acid (A) and methanol (B) with a gradient program of 0−30 min, 40%−100% B linear gradient elution. The injection volume was 20 μL and the flow rate was 1 mL/min. The detection wavelength was set at 280 nm. A Thermo Scientific LCQ Ion-Trap Mass Spectrometer (Thermo Fisher Scientific, San Jose, CA, USA) with electrospray ionization (ESI) at positive mode was used for the HPLC-MS/MS analysis. The conditions were: Dry gas flow, 35 L/min; capillary voltage, 3.5 kV; nebulizer, 30 psi; collision energy, 16 eV; desolvation temperature, 400 °C.

To get purified LEGCG, Waters semi-preparative HPLC systems comprise a 600E pump and a 2998 DAD was used. The separation was carried out using a 19 mm Waters Atlantis OBD-C18 column (10 μm, 250 mm). The mobile phases were 0.1% acetic acid (A) and methanol (B). The gradient program was 0−30 min, 40%−100% B linear gradient elution, at a flow rate of 5 mL/min. The injection volume was 100 μL. The UV absorbance was detected at 280 nm. A Bruker Avance 500 MHz NMR spectrometer (Bruker Biospin GmbH, Rheinstetten, Germany) was carried out for the purified LEGCG to identify its molecular structure.

### 2.4. Determination of Lipophilicity

The lipophilicity of the LEGCG was determined using the method according to the literature [18]. Briefly, the 1-octanol was mixed with deionized water and shaken for 24 h before use. Samples were dissolved in the pre-saturated 1-octanol (10 mL), and then 10 mL of pre-saturated water was added. The mixtures were vortexed and kept for 24 h for separation. The concentration of samples in 1-octanol phase (*C_o_*) and water phase (*C_w_*) were measured at 280 nm. The 1-octanol–water partition coefficient (log *p*) was calculated by the following Equation:log *p* = lg (*C_o_*/*C_w_*).(1)

### 2.5. Cell Culture

Human prostate cancer DU145 cells and normal human prostate epithelial RWPE-1 cells were obtained from the American Type Culture Collection (ATCC, Manassas, VA, USA). DU145 cells were maintained in Dulbecco’s modified Eagle medium supplemented with fetal bovine serum (10%) and penicillin-streptomycin (1%). RWPE-1 cells were maintained in keratinocyte serum-free medium supplied with bovine pituitary extract (50 mg/L), L-glutamine (5%), and epidermal growth factor (5 μg/L). All cells were cultured in a humidified 5% CO_2_ incubator at 37 °C.

### 2.6. Cytotoxicity Assay and Cell Proliferation Analysis

The effects of LEGCG on the proliferation of DU145 cells and RWPE-1 cells were determined using MTT assay, described in the literature [16,19]. Briefly, cells (1 × 10^4^) were seeded into 96-well plates and then treated with different doses of LEGCG for varying periods. After incubation, MTT dye was added and incubated for 3 h. Then, DMSO was added and measured at 570 nm by a microplate spectrophotometer (Molecular Devices).

### 2.7. Apoptosis Analysis and Cell Cycle Analysis by Flow Cytometry

Apoptosis was measured by Annexin V-FITC/PI kit. DU145 cells treated with LEGCG for 24 h were collected. Annexin V-FITC (5 μL) and PI (5 μL) were then added to incubate for 15 min in the dark. For cell cycle analysis, DU145 cells treated with different concentrations of LEGCG for 24 h were collected and resuspended in a 10% formalin solution before overnight fixation at 4 °C. The cell pellets were washed with PBS and stained with PI/RNase Staining Buffer in the dark for 4 h. A total of 10,000 cells of each sample were analyzed by a flow cytometer (BD Biosciences, Franklin Lakes, NJ, USA).

### 2.8. Immunoblot Analysis

DU145 cells were treated with LEGCG for 24 h. Then cell lysates were prepared, and the protein concentrations were determined by the BCA kit as previously described [20]. For immunoblot analyses, proteins were denatured in SDS sample buffer and subjected to an SDS-PAGE gel. Then the separated proteins were transferred onto a nitrocellulose membrane. Membranes were incubated with primary antibody followed by peroxidase-conjugated secondary antibody, probed using ECL, and visualized on autoradiography.

### 2.9. Statistical Analysis

All experiments were carried out in triplicate. Values were expressed as mean ± standard deviation. The significant differences between groups were calculated by Student’s *t*-test using GraphPad Prism 7.0 software (GraphPad Software Inc., San Diego, CA, USA).

## 3. Results and Discussion

### 3.1. Structure Elucidation of LEGCG

In this study, the LEGCG was prepared via reaction with the medium-chain fatty acid of lauric acid. The yield (calculated as LEGCG/EGCG) of the LEGCG was 63.21%. According to the previous study, ethanol showed a high conversion rate during the enzymatic esterification of flavan-3-ol, thus ethanol was chosen as the reaction solvent [17,21].

The crude LEGCG was separated by semi-preparative HPLC. The tandem MS/MS of purified LEGCG is shown in Figure 1A. The molecular ion peak detected showed an m/z at 1187, representing [M + H]^+^ of the LEGCG (C_70_H_106_O_15_, MW 1186). The cleavage of the lauric acid moiety (mass 182) from the fragments m/z 1187, 1005, 823, and 641 led to peaks at m/z 1005, 823, 641, and 459, respectively. Thus, LEGCG was identified as tetralauroyl EGCG.

The EGCG molecule has multiple hydroxyl groups, thus the chemical shifts of LEGCG and EGCG were compared using ^1^H and ^13^C NMR to further determine the esterification position. Compared with EGCG, a downfield shift (Δδ 0.34–0.40) of protons in LEGCG was found for H-2′, H-6′, H-2″, and H-6″, indicating that the esterification occurred in the B-ring and D-ring. As shown in Appendix A, a remarkable downfield shift (Δδ 4.39–5.09) was observed for C-3′, C-5′, C-3″, and C-5″, suggesting these might be the esterification sites. The large upfield shift (Δδ 1.31–2.18) was observed for C-4′ and C-4″, suggesting the presence of hydroxyl groups at C-4′ and C-4″. Thus, tetralauroyl EGCG was identified as 3′,5′,3″,5″-4-O-lauroyl EGCG (Figure 1B).

### 3.2. Lipophilicity of LEGCG

The high hydrophilicity of EGCG is considered as having low bioavailability and is not conducive to application in the fat food system. The structural modification, especially enzymatic esterification, is an effective means to improve lipophilicity. Many hydrophilic bioactive compounds have been lipophilized, including EGCG, cinnamic, caffeic acids, ascorbic acid, genistein [22,23,24,25,26], and expanding their application in a more lipophilic system.

The lipophilicity of LEGCG was evaluated by the log *p*. A higher log *p* value indicates higher lipophilicity of the compound. As expected, the log *p* value of LEGCG (4.63 ± 0.14) was significantly higher than their parent EGCG molecule (0.46 ± 0.02). According to the literature, the incorporation rate of tea polyphenols and lipid bilayers was positively correlated with log *p* value [27]. Enhancement of lipophilicity of LEGCG may increase their bioavailability and the liposome-based drug delivery capability.

### 3.3. Effect of LEGCG on the Proliferation of DU145 Human Prostate Cancer Cells

The MTT assay was used to detect the anti-proliferative potential of LEGCG on DU145 cells and RWPE-1 cells. As shown in Figure 2A, LEGCG increased the inhibition rate on human prostate cancer DU145 cells in a concentration-dependent manner at 12–48 h, within the concentrations of 10–50 μg/mL. For normal human prostate epithelial RWPE-1 cells, the inhibition rate significantly increased at 48 h of treatment (*p* < 0.05) and enhanced slowly at 12 and 24 h (Figure 2B). LEGCG showed only a little higher inhibition rate on DU145 cells at the concentrations of 10–50 μg/mL, for 48 h than 24 h, but it exhibited considerably high cytotoxicity on RWPE-1 cells at the doses of 40–50 μg/mL for 48 h (*p* < 0.05). Hence, treatment with LEGCG at the doses of 10–40 μg/mL within 24 h was selected for further study.

The apoptosis rate of DU145 cells was measured by flow cytometry. As shown in Figure 3, the early apoptotic cells (Annexin V^+^/PI^−^ fraction) were markedly increased, and the necrotic cells (Annexin V^+^/PI^+^ fraction we) were not significantly changed. After treatment with LEGCG at 0, 10, 20, and 40 μg/mL for 24 h, the apoptosis rates of DU145 cells were 8.63%, 12.78%, 25.62%, and 58.51%, respectively. Compared to LEGCG, Gupta et al. found that EGCG treatment at 10, 20, 40, and 80 μg/mL for 48 h led to 13.9%, 19.1%, 42.2%, and 58.1% of apoptotic cells on DU145 cells [28]. And Ravindranath et al. reported that the IC50 of EGCG on DU145 cells was 88.66 μM [29]. All the results indicated that LEGCG had a more excellent anti-proliferation capacity against DU145 prostate cancer cells than EGCG.

### 3.4. LEGCG Induces Cell Cycle Arrest on DU145 Cells

To further determine the mechanism of LEGCG-induced anti-proliferation, the effect of LEGCG on the cell cycle of DU145 cells was measured by flow cytometry. The concentrations of LEGCG at 10, 20, and 40 μg/mL were chosen. After treatment with LEGCG for 24 h, the percentages of cells in sub G1 were increased in a dose-dependent manner (1.81%, 2.30%, 50.91%, and 83.42%, respectively), which again demonstrated that LEGCG induced apoptosis on DU145 cells (Figure 4). Meanwhile, the percentages of cells in the G0/G1 phase were significantly enhanced after treatment with LEGCG at 20–40 μg/mL compared to DMSO treatment (*p* < 0.05). The accumulation of cells in the G0/G1 phase indicated that LEGCG arrested DU145 prostate cancer cells in the G0/G1 phase.

To determine these results, the immunoblot analysis of cyclin D1, CDK4, p21, and p53 were performed. Cyclin D1 is considered an oncogene and frequently overexpressed in many cancers [30]. By binding with and activating CDK4 (partner kinases of cyclin D1), cyclin D1 is known to release transcription factors to advance the cell cycle progression from G1 to S phase. Figure 5 shows that the cyclin D1 and CDK4 levels were strongly decreased by LEGCG at the doses of 20 to 40 μg/mL. P53, a tumor suppressor, inhibits the growth of tumor cells by inducing the transcription of p21, which regulates the cell cycle at the G1 to S phase and induces cell cycle arrest by inhibiting the protein expressions of the cyclin–CDK complex [31,32]. Immunoblot analysis revealed that LEGCG remarkably upregulated expression of p21 and p53 at the doses of 20 to 40 μg/mL (Figure 5). These results indicated that LEGCG induced cell cycle arrest by downregulating cyclin D1 and CDK4 levels induced by p53-dependent p21 activation.

### 3.5. Mechanism Underlying LEGCG-Induced Apoptosis on DU145 Cells

Based on the results of LEGCG-induced apoptosis of DU145 cells, the underlying mechanism was further explored by immunoblot analysis. As a transcription factor, p53 regulates many proteins related to inducing apoptosis, including the pro-apoptotic proteins of the B-cell lymphoma family [33,34]. Bcl-2 is an upstream protein in the apoptotic pathway [35]. Bcl-2 is a potent suppressor of apoptosis and has been found at high levels in many cancers [36,37]. Bax is able to neutralize the anti-apoptotic function of Bcl-2 by forming a heterodimer complex with Bcl-2 [38]. Therefore, the ratio of Bax:Bcl-2 plays a key role in cell apoptosis. LEGCG increased the Bax levels and decreased the Bcl-2 levels on DU145 cells, which led to a significant increase in the ratio of Bax:Bcl-2 in a dose-dependent fashion (*p* < 0.05) (Figure 6).

The mitochondrion is an important participator in apoptosis, and the Bcl-2 family proteins play an essential role in the occurrence of mitochondrial dysfunction. The under-expressing of Bcl-2 in cells induces the release of cytochrome *c* in the cytoplasm and promotes the apoptotic response [39]. To explore the downstream events in LEGCG-induced apoptosis on DU145 cells, the expression of cytosolic cytochrome *c* was investigated. As shown in Figure 6, LEGCG remarkably increased the cytosolic cytochrome *c* levels on DU145 cells, which might be a key event in LEGCG-induced apoptosis.

Apoptosis can be activated by two major mechanisms in most cellular systems: The intrinsic pathway and the extrinsic pathway [40]. Caspases perform a key role in the induction of apoptosis and are triggered by both pathways [41]. The release of cytochrome *c* into the cytoplasm could activate the caspase pathway [42]. To find the underlying mechanism by which LEGCG induces apoptosis, we then assessed the activation of caspase pathways. Figure 6 shows that LEGCG caused a strong dose-dependent cleavage of caspase-9 and cleavage of caspase-3 on DU145 cells, suggesting that caspase activation may be involved in the LEGCG-induced apoptosis.

A proposed signaling pathway for the anti-prostate cancer activity of LEGCG is shown in Appendix A. LEGCG treatment increased the p53 level on DU145 cells. Then, p53 activated the p21 and Bcl-2 family members. Activation of p53-dependent p21 leads to a reduction of the cyclin D1 and CDK4 levels, leading to G0/G1 cell cycle arrest. An increase in the ratio of Bax:Bcl-2 results in the release of cytochrome *c* into the cytoplasm, which activates caspase-9 and caspase-3, leading to caspases cleavage and ultimately to apoptosis.

## 4. Conclusions

In the present study, the esterification of EGCG with lauric acid yielded tetralauroyl EGCG. Its specific structure was confirmed by ^1^H and ^13^C NMR as 3′,5′,3″,5″-4-O-lauroyl EGCG. LEGCG had higher lipophilicity than EGCG. LEGCG possessed an anti-proliferation activity on DU145 cells. The mechanism was confirmed as a p53-mediated induction of apoptosis and cell cycle arrest. It is indicated that LEGCG could be used as a functional food supplement to prevent human prostate cancer. Furthermore, the anti-prostate cancer activities of LEGCG in vivo need further study.

## Figures and Tables

**Figure 1 nutrients-12-00092-f001:**
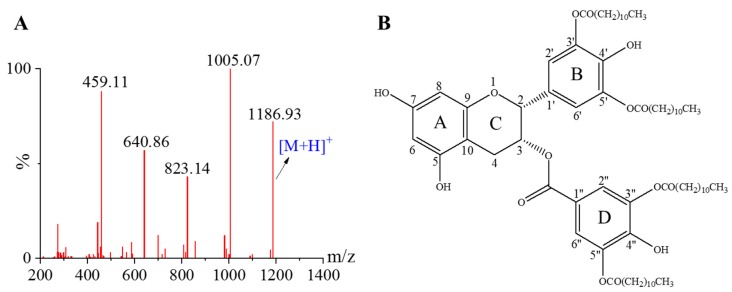
The (**A**) tandem mass spectrometry and (**B**) structure of lipophilized epigallocatechin gallate derivative (LEGCG).

**Figure 2 nutrients-12-00092-f002:**
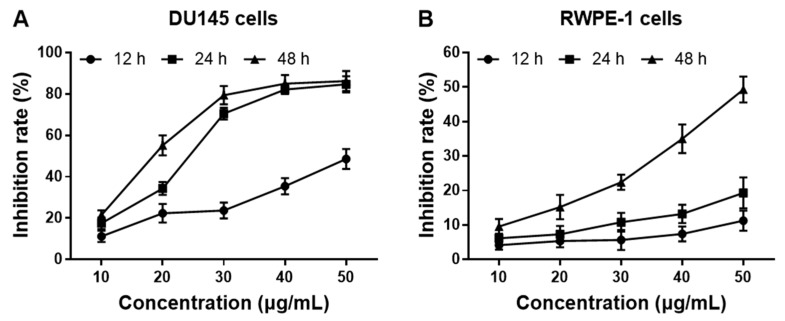
The anti-proliferation effect of LEGCG in (**A**) DU145 cells and (**B**) RWPE-1 cells.

**Figure 3 nutrients-12-00092-f003:**
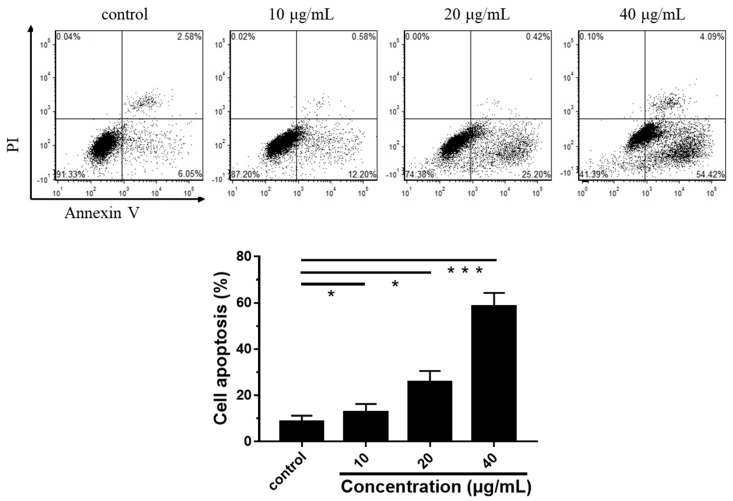
Flow cytometric analysis of cell apoptosis induced by treatment with LEGCG for 24 h on DU145 cells. *, 0.01 < *p* < 0.05, **, 0.001 < *p* < 0.01, ***, *p* < 0.001.

**Figure 4 nutrients-12-00092-f004:**
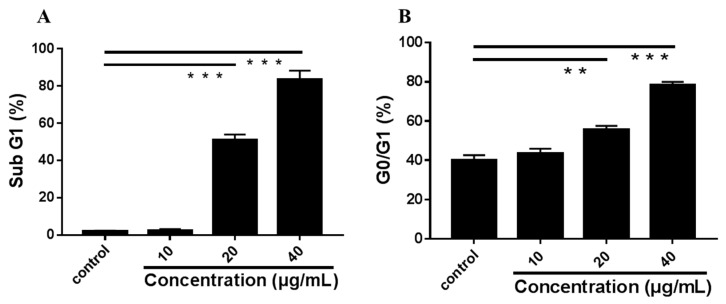
Flow cytometric analysis of cell cycle induced by treatment with LEGCG for 24 h on DU145 cells. (**A**) The percentages of cells in sub G1. (**B**) The percentages of cells in G0/G1 phase. *, 0.01 < *p* < 0.05; **, 0.001 < *p* < 0.01; ***, *p* < 0.001.

**Figure 5 nutrients-12-00092-f005:**
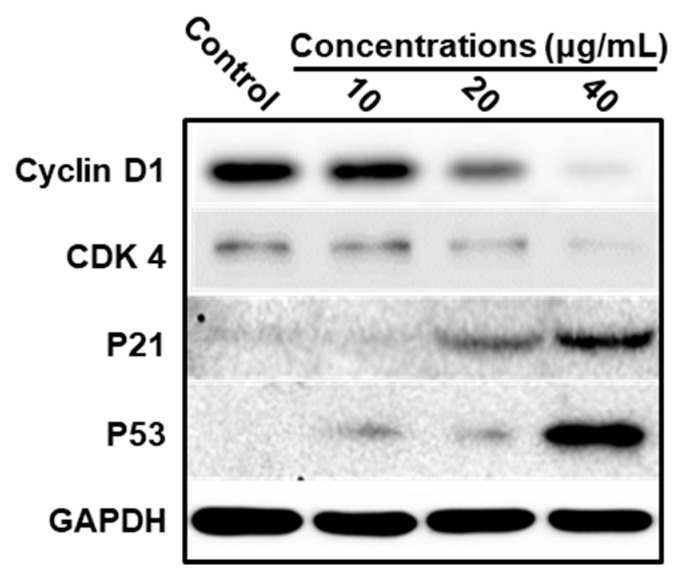
Western immunoblot analysis of the expressions of Cyclin D1, CDK4, P21, and P53 on DU145 cells. Cells were treated with LEGCG for 24 h.

**Figure 6 nutrients-12-00092-f006:**
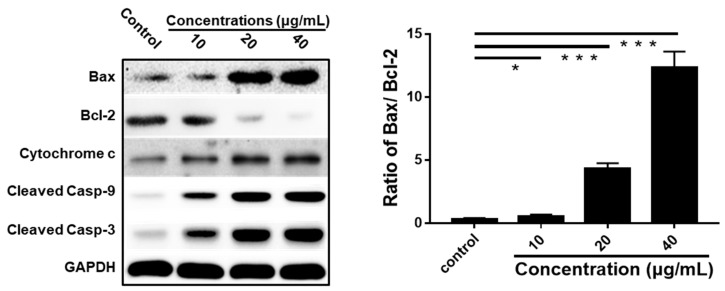
Western immunoblot analysis of the expressions of Bax, Bcl-2, cytochrome *c*, cleaved caspase-9, and cleaved caspase-3 on DU145 cells. Cells were treated with LEGCG for 24 h. *, 0.01 < *p* < 0.05; **, 0.001 < *p* < 0.01; ***, *p* < 0.001.

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
