# Peer review of "Lipophilized Epigallocatechin Gallate Derivative Exerts Anti-Proliferation Efficacy through Induction of Cell Cycle Arrest and Apoptosis on DU145 Human Prostate Cancer Cells"

_nutrients, 2019, doi:10.3390/nu12010092_

Round 1
Reviewer 1 Report
The manuscript “Lipophilized epigallocatechin gallate derivative exerts anti-proliferation efficacy through induction of cell cycle arrest and apoptosis on DU145 human prostate cancer cells” written by Jun Chen et al. has a good experimental design and displays convincing and interesting results. However, background discussion in the introduction section as well as the discussion itself should be improved by citing key papers. Many polyphenol esters have been synthesized and showed in vitro cytotoxicity against cancer cell lines. It is therefore important to discuss about the potential of LEGCG and why this molecule is promising.
General comments:
English language should be carefully revised and edited Figures should be split into Fig. X.a. and Fig X.b. when needed A careful review of the literature cited in the text is needed, e.g. citing the original paper instead of a review. Published synthesis of EGCG esters and related major findings should be discussed
Introduction:
L27: ref. 2 does not discuss about EGCG (derivatives only).
L31-36: “Prostate cancer is a common diagnosed malignancy among males in the United States, and diet is considered one of the most likely environmental risk factors [9]”.
“The lower incidence of prostate cancer in East Asian countries than in Western countries may be attributed to the green tea polyphenols [11]”.
The relationship between prostate cancer, diet and green tea polyphenols is not clear. Plus, the review cited as ref. [9] does not discuss about diet in the United States.
L37: using the word “suppresses” is strong. Maybe use “prevents” instead.
L38-40: a careful review of the literature is needed here. Important papers should appear and be discussed, such as Park et al. (https://doi.org/10.1016/j.bmcl.2004.07.063); Matsumara et al. (https://doi.org/10.1016/j.bbrc.2008.10.128) etc…
L43 : define « HPLC-MS/MS” and “NMR”.
L45 : define “MTT”.
Materials and methods:
Part 2.2.: more details should be added, such as the removal of the enzyme.
L60-66: the HPLC-UV-MS analysis is not clear. Is the HPLC system coupled to the mass spectrometer? If you use collision energy then I guess that you performed MS/MS analyses?
L 62 & 72: space needed between “solution” and “(A)”.
L88-89: define “DMEM”, “FBS”, “KSFM”, “BPE”, “EGF”.
L95: more details are needed in this section: different doses and different time periods.
Part 2.9.: which statistical analyses have been performed? Define “SD”
Results and discussion:
Part 3.1.: it would be interesting if you could add the LEGCG synthesis yield. Did you see any isomers in your mixture (e.g., lauryl groups in positions 4’ or 4”)?
L 120: lauric acid contains 12 carbon atoms, which is not considered as a long-chain fatty acid.
Figure 1: the figure showed here is a tandem mass spectrum, I believe.
L126: giving the exact mass of the [M+H]+ ion and comparing it to the theoretical mass would be a plus.
L143-145: synthesis of EGCG esters has also been published. They should be discussed first.
L146-148: please mention the statistical analyses performed, as well as the p value.
L153-161: this section is linked to the figure 2, adding statistical analyses to the cell inhibition rate would support your findings.
L164: “were not significantly change”: please describe the statistical analyses used.
L177: ref [28] is not appropriate.
L182-184 and Figure 4: statistics needed.
L190-192: I would not say that 10µg/mL of LEGCG strongly decrease cyclin D1 and CDK4 levels.
L206: please define “LGSP”.
L212-215 and Figure 6: statistics needed.
L225: the findings discussed in ref [40] are related to tumour necrosis factor-related apoptosis-inducing ligand (TRAIL)-resistant LNCaP cells.
L237-242: a short section discussing about further studies needed to support the use of LEGCG as a food supplement (e.g., in vivo studies, mode of administration etc…) would be a plus.
References:
Ref [25] should not be written using capital letters
Ref [26] and [27] refer to the same paper.
Reviewer 2 Report
Could the authors include, in the section "Materials and Methods", the specifications of the antibodies used for immunoblots? The authors propose a signaling network for the anti-prostate cancer activity of LEGCG. In particular, the authors show that LEGCG treatment increases the p53 level on DU145 cells.
Is this effect due to a direct or indirect action of LEGCG on p53? Is LEGCG able to bind the androgen receptor as well as EGCG?
Author Response
Response to Reviewer 2 Comments
Point 1: Could the authors include, in the section "Materials and Methods", the specifications of the antibodies used for immunoblots? 

Response 1: According to the reviewer’s comment, we have added the specifications of the antibodies used for immunoblots in the section "Materials and Methods".
 Cyclin D1, CDK 4, p21, p27, p53, Bax, Bcl-2, cytochrome c, cleaved caspase-9, cleaved caspase-3 and GAPDH antibodies were purchased from Cell Signaling Technology (Danvers, MA, USA).
Point 2: The authors propose a signaling network for the anti-prostate cancer activity of LEGCG. In particular, the authors show that LEGCG treatment increases the p53 level on DU145 cells. Is this effect due to a direct or indirect action of LEGCG on p53? Is LEGCG able to bind the androgen receptor as well as EGCG?
Response 2: In this paper, we found LEGCG treatment increased the p53 level on DU145 cells, we think this effect is a direct action of LEGCG on p53. DU145 cell line is an androgen-independent prostate cancer cell line, thus we are not sure if LEGCG is able to bind the androgen receptor as well as EGCG, and we are studying further.

Reviewer 3 Report
This manuscript describes the enzymatic preparation of lipophilized epigallocatechin gallate and its anti-cancer activity. The authors prepared titled compound in a reasonable method and they found this compound showed anti-proliferative effect against prostate DU145 cancer cells. They also described the mechanism of action of anti-cancer activity. The manuscript should be revised because of following reasons.
The authors prepared the lipophilized epigallocatechin gallate. The structure was confirmed by 1H-and 13C NMR. However , the explanation of 13C-NMR data was lacking. Please describe the data of 13C NMR to support the structure. As to mass spectra, it is better to assign the fragmentation. Please check the references. References [7[, [9], and [32] lacked pages. Reference [17] should be "Journal of Agricultural and Food Chemistry". Reference [27] should be "Bioscience, Biotechnology and Biochemistry". As to the authors, "Y. HARA" should be "Y. Hara, T." and "T. NAKAYAMA" should be "T. Nakayama".
Author Response
Response to Reviewer 3 Comments
Point 1: The authors prepared the lipophilized epigallocatechin gallate. The structure was confirmed by 1H-and 13C NMR. However, the explanation of 13C-NMR data was lacking. Please describe the data of 13C NMR to support the structure. 

Response 1: According to the reviewer’s comment, we have added the explanation of 13C-NMR data to support the structure. “The EGCG molecule has multiple hydroxyl groups, thus the chemical shifts of LEGCG and EGCG were compared using 1H and 13C NMR to further determine the esterification position. Compared with EGCG, a downfield shift (Δδ 0.34-0.40) of protons in LEGCG was found for H-2′, H-6′, H-2″ and H-6″, indicating that the esterification was occurred in the B-ring and D-ring. As shown in Supplementary Table 1, a remarkable downfield shift (Δδ 4.39-5.09) observed for C-3′, C-5′, C-3″ and C-5″, suggesting these might be the esterification sites. The large upfield shift (Δδ 1.31-2.18) observed for C-4′ and C-4″, suggesting the presence of hydroxyl groups at C-4′ and C-4″. Based on the combined 1H and 13C NMR, tetralauroyl EGCG was identified as 3′,5′,3″,5″-4-O-lauroyl EGCG.”
Point 2: As to mass spectra, it is better to assign the fragmentation.
Response 2: According to the reviewer’s comment, we have added the description of the mass spectra fragmentation. “The molecular ion peak detected showed an m/z at 1187, representing [M + H]+ of the LEGCG (C70H106O15, MW 1186). The cleavage of the lauric acid moiety (mass 182) from the fragments m/z 1187, 1005, 823 and 641 led to peaks at m/z 1005, 823, 641 and 459, respectively.”
Point 3: Please check the references. References [7[, [9], and [32] lacked pages.
Response 3: We have checked references [7[, [9], and [32], and found that these references have no pages.
Point 4: Reference [17] should be "Journal of Agricultural and Food Chemistry".
Response 4: According to the reviewer’s comment, we have revised reference [17] as followed: “M. Chen, S. Yu, Characterization of Lipophilized Monomeric and Oligomeric Grape Seed Flavan-3-ol Derivatives, Journal of Agricultural and Food Chemistry, 65 (2017) 8875-8883”.
Point 5: Reference [27] should be "Bioscience, Biotechnology and Biochemistry". As to the authors, "Y. HARA" should be "Y. Hara, T." and "T. NAKAYAMA" should be "T. Nakayama".
Response 5: According to the reviewer’s comment, we have revised reference [27] as followed: “T. Hashimoto, S. Kumazawa, F. Nanjo, Y. Hara, T. Nakayama, Interaction of tea catechins with lipid bilayers investigated with liposome systems, Bioscience Biotechnology and Biochemistry, 63 (1999) 2252-2255”

Round 2
Reviewer 1 Report
The authors answered to the different points listed in the comments and suggestion file, and significantly improved their manuscript.
However,
- Figure 1 legend is still wrong. I believe that part A is a tandem mass spectrum (or MS/MS) and not a "mass spectra".
- An effort has been made regarding English-language writing, but careful reading and editing is still required. It seems to me that the authors should call on a native English speaker to help them editing the writing of the manuscript.
